# Role of PNPLA3 in the Assessment and Monitoring of Hepatic Steatosis and Fibrosis in Patients with Chronic Hepatitis C Infection Who Achieved a Sustained Virologic Response

**DOI:** 10.3390/medicina57111153

**Published:** 2021-10-24

**Authors:** Oana Irina Gavril, Lidia Iuliana Arhire, Otilia Gavrilescu, Mihaela Dranga, Oana Barboi, Radu Sebastian Gavril, Roxana Popescu, Cristina Cijevschi Prelipcean, Anca-Victorita Trifan, Catalina Mihai

**Affiliations:** 1Department of Medical Specialties (I), Faculty of Medicine, “Grigore T. Popa” Universityof Medicine and Pharmacy, 700111 Iași, Romania; ir.ungureanu@yahoo.com (O.I.G.); otilia.gavrilescu@umfiasi.ro (O.G.); mihaela.dranga@umfiasi.ro (M.D.); oany_leo@yahoo.com (O.B.); cristinacijevschi@yahoo.com (C.C.P.); ancatrifan@yahoo.com (A.-V.T.); catalinamihai@yahoo.com (C.M.); 2Department of Medical Specialties (II), Faculty of Medicine, “Grigore T. Popa” University of Medicine and Pharmacy, 700111 Iași, Romania; lidia_graur@yahoo.com; 3Department of Medical Genetics, Faculty of Medicine, “Grigore T. Popa” University of Medicine and Pharmacy, 700111 Iași, Romania; roxana.popescu2014@gmail.com

**Keywords:** chronic hepatitis C virus, patatin-like phospholipase domain-containing3, direct acting antiviral, sustained viral response

## Abstract

*Background and Objectives*: Hepatic diseases are an important public health problem. All patients with chronic hepatitis C virus (HCV) infection receive treatment, regardless of hepatic fibrosis severity. However, evaluation of hepatic fibrosis and steatosis is still useful in assessing evolution, prognosis and monitoring of hepatic disease, especially after treatment with direct-acting antivirals (DAAs). The aim of this study was to assess the link between patatin-like phospholipase domain-containing 3 (PNPLA3) polymorphism and the degree of hepatic steatosis and fibrosis in patients with chronic HCV infection, as well as changes in steatosis and fibrosis three monthsafter obtaining a sustained viral response (SVR). *Materials and Methods*:Ourstudy included 100 patients with chronic hepatitis C (CHC) infection and compensated cirrhosis who received DAA treatment and who were evaluated using Fibromax prior to and 3 months after SVR. The influence of PNPLA3 (CC, CG, GG) genotype among these patients on the degree of post-treatment regression of steatosis and fibrosis was assessed. *Results*: Regression was noticed in the degree of both hepatic steatosis and hepatic fibrosis post-DAA treatment (three months after SVR). Analysis of the correlation between PNPLA3 genotype and fibrosis indicated that the average level of fibrosis (F) before DAA treatment was higher in patients with the GG genotype than in patients with the CC or CG genotype. Three months after SVR, the average level of fibrosis decreased; however, it remained significantly increased in GG subjects compared to that in CC or CG patients. The degree of hepatic steatosis before treatment was not significantly different among patients with different PNPLA3 genotypes, and no significant correlations were observed three months after SVR. *Conclusions*: The genetic variants of PNPLA3 influence the evolution of hepatic fibrosis. The GG subtype plays an important role in the degree of hepatic fibrosis both before and after treatment (three months after SVR)and could be a prognostic marker for assessment of post-SVR evolution.

## 1. Introduction

Global estimates suggest that 71 million people present hepatopathies caused by chronic hepatitis C virus (HCV) infection. Direct-acting antiviral (DAA) therapy access has significantly improved the prognosis and evolution of the disease [1].

The availability of DAA treatment worldwide and the simplification of therapeutic strategies determine the possibility of eliminating HCV through improvement of screening methods, identification of infected subjects with active infection and timely treatment initiation.

Considering the efficiency of DAA treatment and the good tolerance to these drugs, elimination (not eradication, in the absence of an efficient vaccine) is possible when taking into account the improvement of screening methods and treatment accessibility. Elimination, as defined by the World Health Organization (WHO), is estimated to be 80% in 2030.

However, although DAA revolutionized HCV treatment and more than 95% of subjects have achieved a sustained viral response (SVR), the prognosis is not clear because hepatocellular carcinoma remains a major risk, along with hepatic disease progression. Evolution post-SVR and hepatocellular carcinoma risk depend on fibrosis and steatosis severity.

HCV has led to the most indications for liver transplantation, although currently, HCV tends to be replaced by toxic hepatic disease, nonalcoholic fatty liver disease (NAFLD) and hepatocellular carcinoma [2].

In 2008, the attention of the scientific community on patatin-like phospholipase domain-containing 3 (PNPLA3) increased due to a cohort-type study that reported a close correlation between a genetic variant of PNPLA3, rs738409 C>G, and an increased risk of developing hepatic steatosis and inflammation. This study was followed by separate confirmations in other cohorts. The PNPLA3 variant is a singular polymorphism leading to a change from a cytosine (C) to a guanine (G), resulting in substitution of an isoleucine (I) with a methionine (M) at position 148 (I148M) [3].PNPLA3 represents a lipid droplet-associated protein that has been shown to have hydrolase activity toward triglycerides and retinyl esters. Reports have shown that PNPLA3 rs738409[G] (148M) variant is associated with hepatic triglyceride accumulation (steatosis), inflammation, fibrosis, cirrhosis, and even hepatocellular carcinoma regardless of etiologies including alcohol- or obesity-related and others.Alcoholic and non-alcoholic liver diseases often begin with simple steatosis and progress to hepatitis, fibrosis/cirrhosis, and even liver cancer. Both environmental and genetic factors contribute to the development of these chronic liver diseases. Among the well documented genes, PNPLA3 has the broad impact on ALD (alcoholic liver disease) and NAFLD [4].

Recent studies have indicated that I148M PNPLA3 influences steatosis and/or fibrosis development in patients with HCV. A 2011 study compared two groups of Italian patients with HCV infection and indicated for the first time that PNPLA3 genetic variants influence steatosis development regardless of age, sex, body mass index, diabetes mellitus, alcohol consumption and viral genotype [5]. In the same study, PNPLA3 promoted fibrosis and the risk of developing hepatocellular carcinoma in patients with HCV.

The aim of the present study was to determine whether PNPLA3 polymorphism is linked to the degree of hepatic steatosis and fibrosis in patients with HCV infection and whether PNPLA3 is related to fibrosis and steatosis evolution three monthspost-SVR.

## 2. Materials and Methods

We conducted a prospective study that included 102 patients with viral hepatitis C infection (both previously known cases of liver disease and newly diagnosed cases)that were evaluated in our clinic in an outpatient–ambulatory setting.Patients were evaluated pretreatment and three months after obtaining SVR based on clinical, biological and imaging criteria(abdominal ultrasound and upper endoscopy) at the Institute of Gastroenterology and Hepatology within the County Emergency Hospital “St. Spiridon”, Iași, Romaniafrom January 2018 to March 2020. The initial visit was before DAA treatment, when genotyping of the PNPLA3 polymorphism was also performed (T0). The second visit was three months post-SVR (T3). During the evaluation according to the SVR protocol, 2 subjects were nonresponders. In our study, we evaluated only patients who obtained SVR, and thus, the study consisted of 100 patients. DAA therapy included ombitasvir/paritaprevir/ritonavir + dasababuvir or ledipasvir + sofosbuvir.

The inclusion criteria (according to the National Health Fund protocols and international guidelines’ recommendations) were as follows: patients with a positive RNA-HCV test, older than 18 years, who signed the informed consent. The exclusion criteria were as follows: patients with nondetectable HCV RNA, subjects who did not receive a favorable recommendation from medical consultants for DAA treatment due to associated comorbidities, subjects with decompensated liver disease defined by the presence of hepatic encephalopathy, variceal gastrointestinal bleeding, ascites or jaundice (as detected by the clinical, biological and imaging exams), subjects diagnosed with hepatocellular carcinoma or other types of malignant tumors, patients with diabetes mellitus, obese subjects (BMI > 29), alcoholconsumers, patients with viral co-infections, and patients with steatogenic medication.

All patients signed a written informed consent for inclusion. The study was conducted in accordance with the Declaration of Helsinki, and the protocol was approved by the Ethics Committee of the “Grigore T. Popa” University of Medicine and Pharmacy in Iasi, Romania.

All patients included in the study had genotype 1b, detectable viremia, a complete liver panel for liver disease, an evaluation of comorbidities and a favorable opinion from the specialist doctor for other conditions, and an evaluation of liver fibrosis via a noninvasive test (Fibromax). The degree of hepatic steatosis and fibrosis was assessed using Fibromax (FibroTest, SteatoTest) upon treatment initiation and 3 months after SVR was achieved. All the enrolled patients were investigated using the same testing method to avoid differences in diagnosis.

Fibromax is a blood test in which the sample is taken in the morning and at least 2 mL of blood is required. The samples were processed via photometry and immunonephelometry. Fibromax includes several parameters represented by 5 noninvasive tests: the FibroTest (assesses the degree of liver fibrosis), ActiTest (determines necroinflammatory activity), SteatoTest (assesses the degree of hepatic steatosis), NashTest (determines the presence of NASH in patients with metabolic syndrome), and AshTest (determines the degree of liver damage in subjects with chronic alcohol consumption).

The results of these 5 parameters are calculated using formulas correlated with the patient’s date of birth, sex, height and weight. A series of blood tests are assessed using this score: alpha 2 macroglobulin, haptoglobin, apolipoprotein A1, total bilirubin, alanine aminotransferase, aspartate aminotransferase, gamma-glutamyltranspeptidase, fasting serum glucose, cholesterol, and triglycerides. In the present study, we took only FibroTest and SteatoTest results into consideration.

According to the calibration used by the local laboratory, the cut-offs of the FibroTest score for delimiting the degrees of liver fibrosis were: F1 ≤ 0.25, 0.25 < F2 ≤ 0.5, 0.5 < F3 ≤ 0.75, and F4 > 0.75. Similarly, the cut-offs of the SteatoTest score for delimiting the degrees of liver steatosis were: S0 ≤ 0.25, 0.25 < S1 ≤ 0.5, 0.5 < S2 ≤ 0.75, and S3> 0.75.

The criteria set down by the National Health Fund were as follows: subjects with fibrosis F2, F3, and F4. Patients with fibrosis F0 and F1 were not eligible to initiate DAA at that time.

In addition, for all patients, blood was collected for genotyping of the PNPLA3 rs738409 polymorphism.

TaqMan-based real-time PCR genotyping was used to detect PNPLA3 polymorphisms, and allele discrimination was performed by evaluating the fluorescence of mutant alleles compared to the fluorescence of wild-type alleles using an XY or scatter plot chart.

Initially, the 40X Predesigned TaqMan^®^ SNP Genotyping Assay (Thermo FisherScientific, Paisley, UK) probe mix was diluted to a 20× solution concentration.

For each patient, 20 nanograms of genomic DNA was amplified (sample volume 4.50 µL). Each working group included 1 negative control to exclude contamination, 2 positive controls (one homozygous for the normal allele and one homozygous for the mutant allele) and samples of patients in the study group. Three genotype standards were analyzed (CC, CG, GG).

For statistical analysis, SPSS version 18.0 was used. Descriptive analysis was conducted using ANOVA. For significant difference calculations, non-parametric chi square and Kruskal–Wallis tests were used to compare two or more intragroup distributions of frequency. In order to be able to apply tests of statistical significancy we assessed the normal range of values. Skewness test values between −2 and 2 and mean close to median values show that the variables were continuous. The statistical indicators according to ANOVA test were: indicators of the mean value (mean, median, modulus, minimum and maximum values, etc.) and indicators of dispersion (standard error, standard deviation, coefficient of variation). For comparisons of continuous variables between groups, Student’s *t* test and a paired samples *t* test were applied to a significance threshold of 95% (*p* < 0.05). For multiple comparisons of normal distributed series of values, a post hoc Bonferroni test was applied after one way ANOVA.

## 3. Results

### 3.1. Baseline Characteristic of the Patients

The study group consisted of 100 patients with chronic hepatitis C infection, of whom 65% were female.

Patient age ranged from 35 to 77 years, with an average age of 60.74 years ± 8.58, close to the group median value (61 years), suggesting homogeneity of the series of values, which was confirmed by the results of Skewness and Kurtosis tests. Thus, statistical significance tests for continuous variables could be applied.

The percentage distribution by sex and age did not show significant differences (*p* = 0.089).

Out of the 100 subjects included in the study, 72 patients received treatment with ombitasvir/paritaprevir/ritonavir + dasababuvir, and 28 subjects received treatment with ledipasvir + sofosbuvir.

Values for fibrosis ranged from 0.32 to 0.96, with an average value of 0.65 ± 0.18, close to the median value of the group (0.65), suggesting homogeneity of the series of values, which was confirmed by the results of Skewness and Kurtosis tests. Thus, statistically significant tests for continuous variables could be applied.

In the current study, most cases presented grade F4 fibrosis (43%).

The values for steatosis ranged from 0.11 to 0.89, with an average level of 0.50 ± 0.18, relatively close to the median value of the group (0.49), suggesting homogeneity of the series of values, which was confirmed by the results of Skewness and Kurtosis tests. Thus, statistically significant tests for continuous variables could be applied.

Most cases were grade S2 steatosis (37%) (Table 1).

The subgroup distribution of the baseline characteristics according to the PNPLA3 genotype, age, BMI, gender, and the degrees of fibrosis and steatosis is illustrated in Table 2. Regarding patients with GG genotype, 61.5% presented F4 fibrosis at baseline compared to patients with CC and CG genotypes which were diagnosed with F4 fibrosis at baseline in 48.3% and 24.1% of cases, respectively.

Before treatment, the mean levels of total bilirubin were significantly increased in GG genotype compared to CC and CG genotypes (1.90 vs. 0.77 and 0.94; *p* = 0.001) and the mean levels of total cholesterol were significantly increased in CC genotype compared to CG and GG genotypes (154.14 vs. 147.62 and 115.33; *p* = 0.019), the others parameters, except aspartate aminotransferase (AST), registering average values slightly lower for the GG genotype than those registered for the CC or CG genotypes (*p* > 0.05).

### 3.2. Evolution of Steatosis and Fibrosis Three Months Post-SVR

Three months post-SVR, a significant decrease was observed in the degree of hepatic steatosis and fibrosis, both globally and within subgroups of distinct steatosis and fibrosis degrees (Table 3).

The degree of fibrosis regressed in 76% of the studied group (progression was seen in 24% of the studied group), and the degree of steatosis regressed in 81% of the group (18% progression, 1% stationary).

### 3.3. Correlation between PNPLA3 Polymorphism, Steatosis and Fibrosis Pre-Treatment (T0)

PNPLA3 genotyping revealed the following: 55% of the subjects presented the CC genotype, 32% of the subjects presented the CG genotype, and 13% of subjects presented the GG genotype.

The distribution of steatosis and fibrosis before SVR correlated with PNPLA3 is presented in Table 4.

Before treatment, the mean levels of fibrosis were significantly increased in GG genotype compared to CC genotype (0.79 vs. 0.65; *p* = 0.042) or CG genotype (0.79 vs. 0.60; *p* = 0.004), while mean values of steatosis did not differ significantly among genotypes.

### 3.4. Correlation between PNPLA3 Polymorphism and Steatosis and Fibrosisthree Monthspost-SVR (T3)

In patients with the CC genotype, the mean levels of both F (fibrosis: 0.65 vs. 0.52; *p* = 0.001) and S (steatosis: 0.51 vs. 0.33; *p* = 0.001) decreased significantly three monthspost-SVR.

In patients with the CG genotype, the mean levels of both F (fibrosis: 0.60 vs. 0.55; *p* = 0.031) and S (steatosis: 0.48 vs. 0.35; *p* = 0.001) decreased significantly three monthspost-SVR.

In patients with the GG genotype, the mean levels of both F (fibrosis: 0.79 vs. 0.70; *p* = 0.006) and S (steatosis: 0.52 vs. 0.37; *p* = 0.005) decreased significantly three monthspost-SVR.

Analysis of the correlation between fibrosis and PNPLA3 led to the following findings (Figure 1):

The mean level of F before treatment was higher in patients with GG (0.79) than in those with CC (0.65) or CG (0.60) (*p* = 0.006).The mean level of fibrosis decreased three monthspost-SVR but remained significantly higher in patients with GG (0.70) than in those with CC (0.50) or CG (0.55) (*p* = 0.004).

The mean level of S pre-treatment did not significantly differ depending on PNPLA3 genotype (0.51; 0.48; 0.52; *p* = 0.738), and the lack of correlation was maintained even three monthspost-SVR(0.33; 0.35; 0.37; *p* = 0.481).

We noticed regression of steatosis severity but no correlation with PNLPA3 polymorphism (Figure 2).

The distribution of steatosis and fibrosis three months post-SVR correlated with PNPLA3 is presented in Table 5.

Correlated with the PNPLA3 polymorphism, fibrosis regression was more frequently observed in 84.6% of GG genotype subjects and 79.3% of CC genotype subjects (*p* = 0.279), and steatosis regression was more frequently observed (82.8%) in CG genotype patients and 81% of CC genotype patients (*p* = 0.554) (Figure 3).

Three months post-SVR fibrosis F4 cases decreased significantly for GG genotype (from 61.5% to 38.5%) and F3 cases increased from 30.8% to 46.2%.

Three months post-SVR steatosis S2 cases decreased significantly for GG genotype (from 53.8% to 23.1%) and S0 cases increased from 0% to 38.5% (Table 6).

## 4. Discussion

Our study assessed liver steatosis and fibrosis regression using a noninvasive method (Fibromax) before and three monthspost-SVR, and to the best of our knowledge, this is the first study of this type to be conducted. Many studies have investigated this relationship by means of transient elastography and liver biopsy, and the latter is still considered the gold standard for assessing the degree of fibrosis and steatosis [6,7].

At the same time, we must state that this is the only study to date to assesses the evolution three monthspost-SVR patients according to PNPLA3 polymorphism.

The vast majority of HCV patients that undergo DAA treatment achieve SVR [8]. However, in a small percentage, the treatment is not effective [9]. Similarly, in the present study, a small percentage of subjects were nonresponders.

The regression of fibrosis three monthspost-SVR is variable and is certainly found in all patients [10]. In our study, regression of liver fibrosis was noted in the vast majority of the subjects (both mild and severe fibrosis cases). The same significant decrease was observed in terms of the degree of hepatic fat accumulation.

In the present study, an important decrease in the degree of hepatic fat accumulation and mainly in the degree of hepatic fibrosis was observed shortly after the end of DAA treatment, at least 3 months post-SVR. The degree of fibrosis regressed in 76% of the patients and progressed in 24%. Hepatic steatosis regressed in 81% of cases, was stationary in 1% and progressed in 18%.

In a study that included 97 patients that obtained SVR, hepatic fibrosis measured via liver biopsy was found to regress in 44 patients (45%), progress in 6 patients (6%) and remain constant in 47 patients (48%). In subjects with progressive fibrosis, the incidence of hepatic complications was much higher than in those who presented regression or constant fibrosis after obtaining SVR (33% vs. 4% in 5 years). If fibrosis regresses, it occurs through a very slow process. The average rate of the fibrosis regression after a period of 3.7 years was—0.28 U/year [11].

A prospective study that included 2326 patients with HCV infection who were assessed 4–6 months after obtaining SVR over a period of at least one year using transient elastography showed fibrosis regression in 27.4% of the patients, while in 50.8% the fibrosis degree remained constant [12].

Another study reported significant liver fibrosis regression in 40% of 260 subjects with HCV infection treated with DAAs, and regression was more significant in subjects with advanced fibrosis and liver cirrhosis during the initial pre-treatment evaluation [13].

Another study that included 304 HC patients treated with a pegylated interferon (IFN-based) (153 patients) or interferon-free DAA (151 patients) used transient elastography to assess fibrosis and found that the percentage of patients with F4 stage decreased from 56.6% to 36.5%. The study was prospective but with a short follow-up period (24 weeks) [14].

There is no clear evolution of the fibrosis progression after viral healing. In our study, progression of liver fibrosis was found in 24% of the subjects, without taking into account other risk factors. A similar study showed that diabetes and obesity are risk factors for progression of liver disease [12].

One of the few studies assessing hepatic steatosis in DAA-treated patients, conducted in 2018, shows the presence of hepatic steatosis in almost half of the subjects post-SVR. Patients were evaluated using transient elastography with controlled attenuation parameters [15].

Liver fibrosis and steatosis regression remain debatable. Some studies have shown that it depends on the assessment method used, and the vast majority of studies assess liver fibrosis by means of transient elastography [16].

In our study, fibrosis and steatosis regression was observed in many patients and in a shorter period of time than observed in other studies. This is likely because the evaluation method we used was Fibromax, which includes biological parameters (alanine aminotransferase, aspartate aminotransferase, gamma-glutamyltranspeptidase), and their values change significantly three monthspost-SVR.

Our study explains, to some extent, the interindividual differences, showing that the degree of liver fibrosis differed significantly depending on PNPLA3 genotype, both before treatment and after three monthspost-SVR.

Previous studies have shown an association between the GG genotype and the degree of liver fibrosis in patients with HCV infection [17,18].

The GG genotype was associated with more severe forms of fibrosis before initiation of DAA treatment than the CC and CG genotypes. This relationship was also observed in the reassessment ofthree monthspost-SVR patients, where the degree of hepatic fibrosis and steatosis regressed, but severe fibrosis was still associated with the GG genotype.

Hepatic steatosis in NAFLD, metabolic syndrome, and nonalcoholic steatohepatitis (NASH) is correlated with PNPLA3 polymorphisms [19,20,21], but in our study, no significant differences associated with genotype were observed. One explanation might be that we did not use the same method used in other studies to show this relationship (percutaneous liver biopsy, abdominal ultrasound). However, some other studies have shown similar results [22].

In a study that included 117 patients treated with DAAs and evaluated using transient elastography for hepatic fibrosis and controlled attenuation parameters for hepatic steatosis, PNPLA3 genotype was not correlated with the degree of hepatic steatosis [20]. Although most studies have shown clear associations between GG genotype and liver fat accumulation, the vast majority have been performed on subjects with NAFLD, where the risk factors pertaining to metabolic syndrome are predominant. In the present study, it seems that the viral component plays an essential role, which in association with multiple risk factors (alcohol, obesity, diabetes) prevents observation of a significant association between genotype and severity of hepatic steatosis. Our study did not evaluate parameters (such as obesity, dyslipidemia, and diabetes mellitus) other than HCV that might influence steatosis. Although genotype 1 is not as steatogenic as genotype 3, it seems that HCV can cause steatosis frequently. The high incidence of steatosis in our study group might be explained by other factors.

The degree of liver steatosis and fibrosis decreases three monthspost-SVR, regardless of genotype. It seems that the gene does not intervene in regression, but the GG genotype has been shown to be associated with severe fibrosis both before and three monthspost-SVR, which was not the case with hepatic steatosis. We explain this aspect by the fact that fibrosis is a more accurate marker of viral liver disease, while steatosis is influenced by other factors, which were not evaluated in the present study.

Our study has a series of limitations that must be mentioned. One of these is the limited number of patients and the short monitoring period (9 months). In addition, another limitation that should be mentioned is that the severity of the liver fibrosis and steatosis was evaluated using a noninvasive method and not by direct histological assessment. Moreover, other parameters that could have influenced liver steatosis and fibrosis were not taken into account.

## 5. Conclusions

A significant decrease in hepatic fibrosis and steatosis degree is seen in HCV-infected subjects three monthspost-SVR.

In our study, an association between PNPLA3 gene polymorphisms in HCV and liver fibrosis was observed, but there were no significant correlations with hepatic steatosis.

Regression three monthspost-SVRof liver fibrosis and steatosis is not influenced by genotype, although the GG genotype is associated with the highest degree of fibrosis both before and after SVR.

Further studies are needed to elucidate the genetic role of PNPLA3 in the severity of liver disease and its relationship with post-SVR evolution.

## Figures and Tables

**Figure 1 medicina-57-01153-f001:**
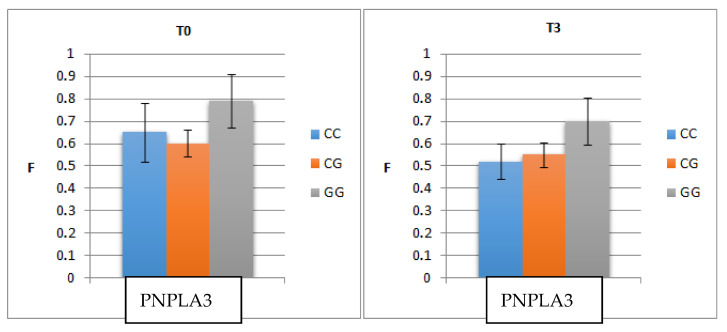
Correlation of pre- and post-treatment fibrosis (three months post-SVR)with PNPLA3 genotype. T0—the initial visit; T3—the second visit; PNPLA3—patatin-like phospholipase domain-containing3; F—fibrosis.

**Figure 2 medicina-57-01153-f002:**
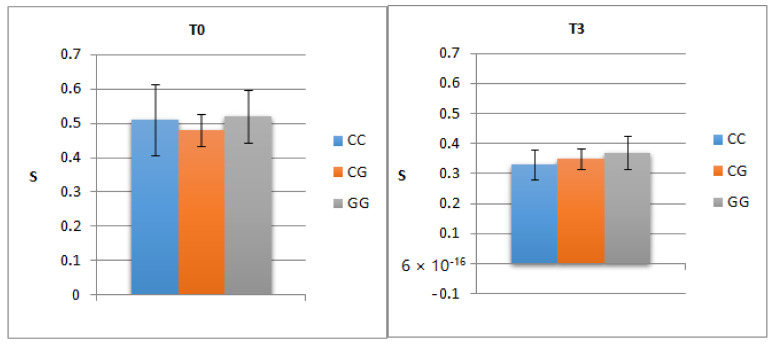
Correlation of pre- and post-treatment steatosis (three months post-SVR) with PNPLA3 genotype.T0—the initial visit; T3—the second visit; PNPLA3—patatin-like phospholipase domain-containing3; S—steatosis.

**Figure 3 medicina-57-01153-f003:**
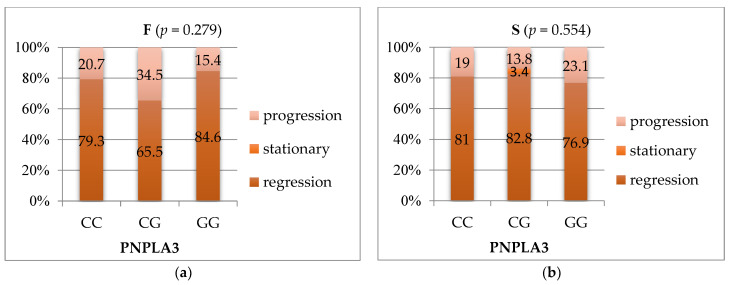
(**a**) Correlation three monthspost-SVR evolution of fibrosis with PNPLA3 genotype; (**b**) Correlation three monthspost-SVRevolution steatosis with PNPLA3 genotype. F—fibrosis; S—steatosis; PNPLA3—patatin-like phospholipase domain-containing3.

**Table 1 medicina-57-01153-t001:** Baseline characteristicsof the entire study population.

Parameter	Value
**Gender**	
Male (%)	35.0
Female (%)	65.0
**Age**	
Average ± SD	60.74 ± 8.58
Min–max/median/Skewness test values	35–77/61/−0.512
**Fibrosis**	
Average ± SD	0.65± 0.18
Min–max/median/Skewness test values	0.32–0.96/0.65/−0.080
**Degree of liver fibrosis**	
F2 (%)	36.0
F3 (%)	21.0
F4 (%)	43.0
**Steatosis**	
Average ± SD	0.50± 0.18
Min–max/median/Skewness test values	0.11–0.89/0.49/−0.015
**Degree of liver steatosis**	
S0 (%)	12.0
S1 (%)	32.0
S2 (%)	37.0
S3 (%)	19.0

F—fibrosis; S—steatosis.

**Table 2 medicina-57-01153-t002:** Baseline characteristics according to PNPLA3 genotype.

Parameters	CC (*n* = 58)	CG (*n* = 29)	GG (*n* = 13)	Test Statistics	*p*
Age					0.85
Average ± SD	61.16 ± 9.66	60.24 ± 6.61	60.00 ± 7.72	F ANOVA test	
BMI					0.548
Average ± SD	27.52 ± 4.74	26.43 ± 3.23	27.67 ± 6.50	F ANOVA test	
Gender					0.896
Male (%)	20 (34.5%)	11 (37.9%)	4 (30.8%)	Kruskal–Wallistests	
Female (%)	38 (65.5%)	18 (62.1%)	9 (69.2%)		
Fibrosis					0.024
F2 (%)	20 (34.5%)	15 (51.7%)	1 (7.7%)	Kruskal–Wallistests	
F3 (%)	10 (17.2%)	7 (24.1%)	4 (30.8%)		
F4 (%)	28 (48.3%)	7 (24.1%)	8 (61.5%)		
Steatosis					0.013
S0 (%)	9 (15.5%)	3 (10.3%)	0 (0.0%)	Kruskal–Wallistests	
S1 (%)	19 (32.8%)	8 (27.6%)	5 (38.5%)		
S2 (%)	14 (24.1%)	16 (55.2%)	7 (53.8%)		
S3 (%)	16 (27.6%)	2 (6.9%)	1 (7.7%)		
ALT					0.878
Average ± SD	94.50 ± 64.24	87.92 ± 63.18	82.17 ± 51.02	F ANOVA test	
AST					0.422
Average ± SD	69.02 ± 36.66	68.69 ± 44.00	91.00 ± 39.82	F ANOVA test	
GammaGT					0.286
Average ± SD	63.67 ± 42.59	66.00 ± 49.69	35.00 ± 16.81	F ANOVA test	
ALP					0.114
Average ± SD	84.86 ± 22.55	103.80 ± 39.70	89.69 ± 11.50	F ANOVA test	
Totalbilirubin					0.001
Average ± SD	0.77 ± 0.33	0.94 ± 0.63	1.90 ± 1.29	F ANOVA test	
ApolipoproteinA1					0.129
Average ± SD	1.43 ± 0.29	1.36 ± 0.27	1.18 ± 0.29	F ANOVA test	
Alfa-2-macroglobulin					0.072
Average ± SD	3.52 ± 0.56	3.18 ± 0.70	3.00 ± 0.85	F ANOVA test	
Heptoglobulin					0.08
Average ± SD	0.72 ± 0.34	0.85 ± 0.57	0.40 ± 0.29	F ANOVA test	
TotalCholesterol					0.019
Average ± SD	154.14 ± 27.78	147.62 ± 35.13	115.33 ± 34.72	F ANOVA test	
Triglicerides					0.689
Average ± SD	90.49 ± 32.70	97.15 ± 24.37	85.50 ± 18.64	F ANOVA test	
Fasting glucose					0.094
Average ± SD	99.54 ± 9.94	97.85 ± 9.56	90.00 ± 9.42	F ANOVA test	

F—fibrosis; S, steatosis; BMI—body mass index; ALT—alanine aminotransferase; AST—aspartate aminotransferase; GGT—gamma-glutamyl transferase; ALP—alkaline phosphatase.

**Table 3 medicina-57-01153-t003:** Evolution of fibrosis and steatosis three months post-SVR.

	T0	T3	Paired Samples Statistics
Fibrosis F2–F4	0.66 ± 0.18	0.55 ± 0.18	0.001
F2	0.45 ± 0.07	0.43 ± 0.13	0.205
F3	0.65 ± 0.04	0.51 ± 0.09	0.001
F4	0.82 ± 0.08	0.67 ± 0.16	0.001
Steatosis S0–S3	0.50 ± 0.18	0.34 ± 0.14	0.001
S0	0.33 ± 0.14	0.30 ± 0.14	0.481
S1	0.34 ± 0.08	0.25 ± 0.11	0.001
S2	0.58 ± 0.06	0.40 ± 0.12	0.001
S3	0.76 ± 0.05	0.45 ± 0.11	0.001

SVR—sustained viral response; T0—initial visit; T3—second visit; F—fibrosis; S—steatosis.

**Table 4 medicina-57-01153-t004:** Mean±SD values of steatosis and fibrosis before treatment according to PNPLA3 genotype (Bonferroni post hoc test pre-treatment).

	F Pre-Treatment			S Pre-Treatment		
	CC	CG	GG	CC	CG	GG
mean ±SD	0.65 ± 0.18	0.60 ± 0.17	0.79 ± 0.12	0.51 ± 0.20	0.48 ± 0.17	0.52 ± 0.16
CC	-	*p* = 0.459	*p* = 0.042	-	*p* = 0.999	*p* = 0.999
CG	*p* = 0.459	-	*p* = 0.004	*p* = 0.999	-	*p* = 0.999
GG	*p* = 0.042	*p* = 0.004	-	*p* = 0.999	*p* = 0.999	-

F—fibrosis; S—steatosis.

**Table 5 medicina-57-01153-t005:** Mean ±SD valuesof steatosis and fibrosis after treatment(three months post-SVR) according to PNPLA3 genotype(Bonferroni post hoc test post-SVR).

	F Three Monthspost-SVR			S Three Monthspost-SVR		
	CC	CG	GG	CC	CG	GG
mean ±SD	0.52 ± 0.17 ^(a)^	0.55 ± 0.18 ^(b)^	0.70 ± 0.15 ^(b)^	0.33 ± 0.14 ^(a)^	0.35 ± 0.13 ^(a)^	0.37 ± 0.18 ^(b)^
CC	-	*p* = 0.999	*p* = 0.003	-	*p* = 0.999	*p* = 0.916
CG	*p* = 0.999	-	*p* = 0.027	*p* = 0.999	-	*p* = 0.999
GG	*p* = 0.003	*p* = 0.027	-	*p* = 0.916	*p* = 0.999	-

F—fibrosis; S—steatosis. (a) *p* <0.001 for paired samples *t*-test pre- vs. post-SVR. (b) *p* <0.05 for paired samples *t*-test pre- vs. post-SVR.

**Table 6 medicina-57-01153-t006:** Post-treatment fibrosis and steatosis according to PNPLA3 genotype three months post-SVR.

Parameters	CC (*n*=58)	CG (*n*=29)	GG (*n*=13)	Test Statistics	*p*
Fibrosis					0.050
F1 (%)	5 (8.6%)	2 (6.9%)	0 (0.0%)	Kruskal–Wallistests	
F2 (%)	35 (60.3%)	15 (51.7%)	2 (15.4%)		
F3 (%)	11 (19.0%)	8 (27.6%)	6 (46.2%)		
F4 (%)	7 (12.1%)	4 (13.8%)	5 (38.5%)		
Steatosis					0.035
S0 (%)	23 (39.7%)	9 (31.0%)	5 (38.5%)	Kruskal–Wallistests	
S1 (%)	27 (46.6%)	16 (55.2%)	4 (30.8%)		
S2 (%)	8 (13.8%)	4 (13.8%)	3 (23.1%)		
S3 (%)	0 (0.0%)	0 (0.0%)	1 (7.7%)		

F—fibrosis; S—steatosis.

## Data Availability

The data presented in this study are available on request from the corresponding author.

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
