# Peer review of "Role of PNPLA3 in the Assessment and Monitoring of Hepatic Steatosis and Fibrosis in Patients with Chronic Hepatitis C Infection Who Achieved a Sustained Virologic Response"

_medicina, 2021, doi:10.3390/medicina57111153_

Round 1
Reviewer 1 Report
The Authors explore the role of genetic susceptibility by pnpla3 polymorphisms to assess the virological response after treatment with DAA in a cohort of compensated, naive HCV patients.
A number of 100 individuals is quite small to allow for a strong evaluation of genetic susceptibility. Fibrosis regression is evaluated through a blood-based tool (Fibrotest). 3 month follow up looks a rather small interval from antiviral therapy. The serum variables change quickly after viral eradication, potentially contributing to the explanation of the results.
Why did the Authors not compare the results with other non-invasive fibrosis tools (APRI, FIB-4)? Fibrosis measured by other techniques (e.g. liver stiffness) improves much more slowly over time.
Some observations and suggestions are found in the following lines:
- I would remove the "new concept" from the title: the evaluation of pnpla3 in post svr with regard to steatosis and fibrosis has been investigated
- In the introduction, the role of pnpla3 should be explained, in order to clarify the association with steatosis and fibrosis. Adiponutrin is involved in the intrahepatic fat export and gene variants allow for an increased rate of liver fat. This is found in different aetiologies, starting from NAFLD and AFLD, to get to HCV infection. These aspects would strengthen the scientific background.
- Materials and methods, line 94: hospitalized for what reason? Patients do not have decompensated liver disease per inclusion criteria. Line 95: imaging criteria? They are not reported in the result sessions.
- Line 108: please define decompensated liver disease: is portal hypertension included in the exclusion criteria? We know that fibrosis regresses even from F4, but the onset of portal hypertesion represents a further worsening of liver disease which makes fibrosis regression quite challenging.
- Line 114 and following are redundant, please move above with the inclusion criteria.
- I would suggest to write the cut-offs for fibrosis and steatosis given by Fibromax: this would allow the reader to better interpret the following results. In addition, in tables a conversion of fibrosis and steatosis values into categorical variables (F0, F1, F2, F3, F4 and S0, S1, S2, S3) would make the association with pnpla3 genotype much clearer.
- Please add the correlation tests in the statistical section. Why adding min-max, in addition to SD, which is already the measure of dispersion?
- Line 167: please write the number of patients, in place of "the rest"
- Table 1: this represents one major point of concern with regard to the plausibility of the results: other characteristics should be given, as pnpla3 impacts on many features of liver disease. How many patients were obese, or had type 2 diabetes? How many consumed alcohol? How many took steatogenic medications? Fibrosis level before ad after treatment might have been linked to multiple reasons,. A multivariable analysis/logistic regression on pnpla3 would have strengthened the evidence of the impact of genotype on fibrosis/steatosis.
- A subrgoup analysis here would better clarify the influence of gene variants: a fibrosis regression from F4 is much more difficult to achieve than in F3 or lower, and this might depend on multiple reasons, including gene susceptibility. How many F4 patients achieved fibrosis regression? What was their genotype? In this setting, is liver steatosis only explained by HCV? Genotype 1 is not as steatogenic as genotype 3, are there other factors determining steatosis in these patients? This is a strong limitation of the study: pnpla3 is mainly related to steatosis, favoring then inflammation and thus fibrosis: this point deserves a greater attention, in my opinion.
- No imaging available for F4 patients? How was the diagnosis made? The use of Fibromax alone would not give the appropriate certaintly that is required in this subgroup of patients.
Author Response
Dear Editor,
We are grateful to the Editorial Board for considering a revised version of our manuscript entitled “Role of PNPLA3 in the Assessment and Monitoring of Hepatic Steatosis and Fibrosis in Patients with Chronic Hepatitis C Infection Who Achieved a Sustained Virologic Response”in the Medicina.
We would like to thank the reviewers for their valuable time and useful contribution. We much appreciate their input, which helped improve our manuscript.
Also, we look forward to hearing from you regarding our submission. We would be glad to respond to any further questions and comments that you may have.
Please find below a detailed point-by-point reply to the comments made by reviewers.
Sincerely,
--
Oana Irina Gavril, MD, PhD
University of Medicine and Pharmacy “Gr. T. Popa” - Iasi - Romania
On behalf of the authors
Letter to the Editor
Reviewers’ Comments to Authors:
Reviewer: 1
The Authors explore the role of genetic susceptibility by pnpla3 polymorphisms to assess the virological response after treatment with DAA in a cohort of compensated, naive HCV patients.
Answer 0.
We would like to thank the Reviewer and the Editorial board for all the remarks regarding our work. We assure the EIC that we have read carefully every suggestion from this decision letter and tried our best to improve the quality of the document accordingly.
Q1. A number of 100 individuals is quite small to allow for a strong evaluation of genetic susceptibility.
Answer1:
Thank you for this important point. Indeed, a larger population would have positively impacted the quality of our paper.
However, there are studies on similar topics published in important journals with good impact factor (IF), on similar population size.
The limited number of patients in our researchis the direct consequence of the limited financial resources that were allocated to the study by our Medical University. In this context, we included the issue of the small-size population in the Limitations section at the end of the Discussions.
Q2. Fibrosis regression is evaluated through a blood-based tool (Fibrotest). 3 month follow up looks a rather small interval from antiviral therapy. The serum variables change quickly after viral eradication, potentially contributing to the explanation of the results.
Why did the Authors not compare the results with other non-invasive fibrosis tools (APRI, FIB-4)? Fibrosis measured by other techniques (e.g. liver stiffness) improves much more slowly over time.
Answer2:
Thank you for this point.
We have chosen to use the Fibrotest non-invasive method as it is accepted and validated by all international guidelines (https://www.journal-of-hepatology.eu/article/S0168-8278(15)00259-7/pdf, https://aasldpubs.onlinelibrary.wiley.com/doi/pdf/10.1002/hep.31060) and is also included in the protocols of our National Health Insurance Fund for assessing fibrosis in the hepatitis C patients.
Moreover, according to the latest meta-analysis comparing fibrosis biomarkers with biopsy in hepatitis C and B patients (https://pubmed.ncbi.nlm.nih.gov/26516104/), Fibrotest was reported to have higher performance than other non-invasive tools: “APRI had lower performances than FIB-4, transient elastography (TE) and FibroTest. TE had lower performance than FibroTest for identifying advanced fibrosis.”
Q3. I would remove the "new concept" from the title: the evaluation of pnpla3 in post svr with regard to steatosis and fibrosis has been investigated
Answer3:
Thank you for this observation.
We updated the title according to the Reviewer’s suggestion, as follows:
“Role of PNPLA3 in the Assessment and Monitoring of Hepatic Steatosis and Fibrosis in Patients with Chronic Hepatitis C Infection Who Achieved a Sustained Virologic Response”
Q4. In the introduction, the role of pnpla3 should be explained, in order to clarify the association with steatosis and fibrosis. Adiponutrin is involved in the intrahepatic fat export and gene variants allow for an increased rate of liver fat. This is found in different aetiologies, starting from NAFLD and AFLD, to get to HCV infection. These aspects would strengthen the scientific background..
Answer4:
Thank you for this extremely well-pointed remark.
We added the following paragraph in the “Introduction” section, quote:
“PNPLA3 represents a lipid droplet-associated protein that has been shown to have hy-drolase activity toward triglycerides and retinyl esters. Reports have shown that PNPLA3 rs738409[G] (148M) variant is associated with hepatic triglyceride accumulation (steato-sis), inflammation, fibrosis, cirrhosis, and even hepatocellular carcinoma regardless of etiologies including alcohol- or obesity-related and others.Alcoholic and non-alcoholic liver diseases often begin with simple steatosis and progress to hepatitis, fibrosis/cirrhosis, and even liver cancer. Both environmental and genetic factors contribute to the development of these chronic liver diseases. Among the well documented genes, PNPLA3 has the broad impact on ALD and NAFLD.”
Q5. Materials and methods, line 94: hospitalized for what reason? Patients do not have decompensated liver disease per inclusion criteria.
Answer5:
We thank the Reviewerfor this very fineobservation!
The patients were indeed not hospitalized in an inpatient setting, but rather all patients were evaluated in an ambulatory setting with dayhospitalizations and no overnight stays.
We modified the first phrase of the “Materials and Methods” to clarify this aspect:
“We conducted a prospective study that included 102 patients with viral hepatitis C infection (both previously known cases of liver disease and newly diagnosed cases) that were evaluated in our Clinic in an outpatient - ambulatory setting.”
Q6. Line 95: imaging criteria? They are not reported in the result sessions.
Answer6:
Thank you for another good observation!
For a more clear point of reference, we defined the imaging criteria by “abdominal ultrasound and upper endoscopy” in the “Materials and Methods” section, as follows (abdominal ultrasound and upper endoscopy were used to diagnose decompensated liver disease; imaging tests are not reported in the results section as all patients with decompensated liver disease were excluded):
“Patients were evaluated pretreatment and three months after obtaining SVR based on clinical, biological and imaging criteria (abdominal ultrasound and upper endoscopy) at the Institute of Gastroenterology and Hepatology within the County Emergency Hospital "St. Spiridon”, Iași, from January 2018 to March 2020.”
“The exclusion criteria were as follows: patients with nondetectable HCV RNA, subjects who did not receive a favorable recommendation from medical consultants for DAA treatment due to associated comorbidities, subjects with decompensated liver disease defined by the presence of hepatic encephalopathy, variceal gastrointestinal bleeding, ascites or jaundice (as detected by the clinical, biological and imaging exams), and subjects diagnosed with hepatocellular carcinoma or other types of malignant tumors.”
Q7. Line 108: please define decompensated liver disease: is portal hypertension included in the exclusion criteria? We know that fibrosis regresses even from F4, but the onset of portal hypertesion represents a further worsening of liver disease which makes fibrosis regression quite challenging.
Answer 7:
Thank you!
We defined decompensated liver disease by the presence of hepatic encephalopathy, varicealgastrointestinal bleeding, ascites or jaundice and updated the text in the manuscript accordingly:
“The exclusion criteria were as follows: patients with nondetectable HCV RNA, subjects who did not receive a favorable recommendation from medical consultants for DAA treatment due to associated comorbidities, subjects with decompensated liver disease defined by the presence of hepatic encephalopathy, variceal gastrointestinal bleeding, ascites or jaundice (as detected by the clinical, biological and imaging exams), subjects diagnosed with hepatocellular carcinoma or other types of malignant tumors, patients with diabetes mellitus, obese subjects (BMI > 29), alcohol consumers, patients with viral co-infections, and patients with steatogenic medication.”
Portal hypertension did not represent an inclusion or exclusion criterion. Our research did not study the regression of portal hypertension but only the regression of liver fibrosis and steatosis.
Q8. Line 114 and following are redundant, please move above with the inclusion criteria.
Answer 8:
Thank you again for a good point!
We removed the redundant parts according to the Reviewer’s suggestion, as follows:
“The inclusion criteria (according to the National Health Fund protocols and international guidelines’ recommendations)were as follows: patients with chronic HCV infection previously known or diagnosed during the current presentation regardless of the degree of hepatic disease, patients with a positive RNA-HCV test, patientsolder than 18 years, and patients who signed the informed consent form. The exclusion criteria were as follows: patients with nondetectable HCV RNA, subjects who did not receive a favorable recommendation from medical consultants for DAA treatment due to associated comorbidities, subjects with decompensated liver disease defined by the presence of hepatic encephalopathy, variceal gastrointestinal bleeding, ascites or jaundice (as detected by the clinical, biological and imaging exams), and subjects diagnosed with hepatocellular carcinoma or other types of malignant tumors, patients with diabetes mellitus, obese subjects (BMI > 29), alcohol consumers, patients with viral co-infections, and patients with steatogenic medication.”
“The eligibility of the included patients was assessed by means of the criteria set down by the National Health Fund recommended by international guidelines: adults, naive subjects or patients with HCV infection.”
Q9. I would suggest to write the cut-offs for fibrosis and steatosis given by Fibromax: this would allow the reader to better interpret the following results. In addition, in tables a conversion of fibrosis and steatosis values into categorical variables (F0, F1, F2, F3, F4 and S0, S1, S2, S3) would make the association with pnpla3 genotype much clearer.
Answer 9:
Thank you for this useful remark!
We added in the “Materials and Methods” section, the cut-off values that delimit the fibrosis and steatosis degrees, as suggested by the Reviewer:
“According to the calibration used by the local laboratory, the cut-offs of the FibroTest score for delimiting the degrees of liver fibrosis were: F1 ≤ 0.25, 0.25 < F2 ≤ 0.5, 0.5 < F3 ≤ 0.75, and F4 > 0.75. Similarly, the cut-offs of the SteatoTest score for delimiting the degrees of liver steatosis were: S0 ≤ 0.25, 0.25 < S1 ≤ 0.5, 0.5 < S2 ≤ 0.75, and S3 > 0.75.”
Q10. Please add the correlation tests in the statistical section. Why adding min-max, in addition to SD, which is already the measure of dispersion?
Answer 10:
Thank you!
In the “Materials and Methods” section, we addedthe correlation tests and the rationale for adding min-max in addition to SD, quote:
“For statistical analysis, SPSS version 18.0 was used. Descriptive analysis was conducted using ANOVA. For significant difference calculations, non-parametric chi square and Kruskal-Wallis tests were used to compare two or more intragroup distributions of frequency.In order to be able to apply tests of statistical significancy we assessed the normal range of values. Skewness test values between -2 and 2 and mean close to median values show that the variables were continuous. The statistical indicators according to ANOVA test were: indicators of the mean value (mean, median, modulus, minimum and maximum values, etc.) and indicators of dispersion (standard error, standard deviation, coefficient of variation).For comparisons of continuous variables between groups, Student’s t test and a paired samples t test were applied to a significance threshold of 95% (P<0.05).For multiple comparisons of normal distributed series of values, a post hoc Bonferroni test was applied after one way ANOVA.”
Q11. Line 167: please write the number of patients, in place of "the rest".
Answer 11:
We updated the text according to the Reviewer’s suggestion. The following parapragh:
“Out of the 100 subjects included in the study, 72 patients received treatment with ombitasvir/paritaprevir/ritonavir + dasababuvir, and the rest received treatment with ledipasvir + sofosbuvir.”
…was replaced by:
“Out of the 100 subjects included in the study, 72 patients received treatment with ombitasvir/paritaprevir/ritonavir + dasababuvir, and 28 subjects received treatment with ledipasvir + sofosbuvir.”
Q12. Table 1: this represents one major point of concern with regard to the plausibility of the results: other characteristics should be given, as pnpla3 impacts on many features of liver disease. How many patients were obese, or had type 2 diabetes? How many consumed alcohol? How many took steatogenic medications? Fibrosis level before and after treatment might have been linked to multiple reasons,. A multivariable analysis/logistic regression on pnpla3 would have strengthened the evidence of the impact of genotype on fibrosis/steatosis.
Answer 12:
Thank you for pointing this out!
The Reviewer is right. For this reason, in order to not influence the results, we excluded patients with diabetes mellitus, obese subjects (BMI>29), consumers of alcohol, patients with viral co-infections and patients with steatogenic medication.
We clarified this aspect in the “Materials and Methods” section, at the exclusion criteria, as follows:
“The exclusion criteria were as follows: patients with nondetectable HCV RNA, subjects who did not receive a favorable recommendation from medical consultants for DAA treatment due to associated comorbidities, subjects with decompensated liver disease defined by the presence of hepatic encephalopathy, variceal gastrointestinal bleeding, ascites or jaundice (as detected by the clinical, biological and imaging exams), subjects diagnosed with hepatocellular carcinoma or other types of malignant tumors, patients with diabetes mellitus, obese subjects (BMI > 29), alcohol consumers, patients with viral co-infections, and patients with steatogenic medication.”
Q13. A subgroup analysis here would better clarify the influence of gene variants: a fibrosis regression from F4 is much more difficult to achieve than in F3 or lower, and this might depend on multiple reasons, including gene susceptibility. How many F4 patients achieved fibrosis regression? What was their genotype?
Answer 13:
Great point!
Following the reviewer’s suggestion, we added the following paragraphs and tables in the “Results” section:
“The subgroup distribution of the baseline characteristics according to the PNPLA3 genotype and the degrees of fibrosis and steatosis is illustrated in Table 2. Regarding patients with GG genotype, 61.5% presented F4 fibrosis at baseline compared to patients with CC and CG genotypes which were diagnosed with F4 fibrosis at baseline in 48.3% and 24.1% of cases, respectively.”
Table 2. Baseline characteristics according to PNPLA3 genotype.
Parameters |
CC (n=58) |
CG (n=29) |
GG (n=13) |
Test statistics |
p |
Age Average ± SD |
61.16±9.66 |
60.24±6.61 |
60.00±7.72 |
F ANOVA test |
0.850 |
BMI Average ± SD |
27.52±4.74 |
26.43±3.23 |
27.67±6.50 |
F ANOVA test |
0.548 |
Gender Male (%) Female (%) |
20 (34.5%) 38 (65.5%) |
11 (37.9%) 18 (62.1%) |
4 (30.8%) 9 (69.2%) |
Kruskal-Wallistests |
0.896 |
Fibrosis F2 (%) F3 (%) F4 (%) |
20 (34.5%) 10 (17.2%) 28 (48.3%) |
15 (51.7%) 7 (24.1%) 7 (24.1%) |
1 (7.7%) 4 (30.8%) 8 (61.5%) |
Kruskal-Wallistests |
0.024 |
Steatosis S0 (%) S1 (%) S2 (%) S3 (%) |
9 (15.5%) 19 (32.8%) 14 (24.1%) 16 (27.6%) |
3 (10.3%) 8 (27.6%) 16 (55.2%) 2 (6.9%) |
0 (0.0%) 5 (38.5%) 7 (53.8%) 1 (7.7%) |
Kruskal-Wallistests |
0.013 |
“Three months post-SVR, a significant decrease was observed in the degree of hepatic steatosis and fibrosis, both globally and within subgroups of distinct steatosis and fibrosis degrees (Table 3).”
Table 3. Evolution of fibrosis and steatosis three months post-SVR
|
T0 |
T3 |
Paired samples statistics |
Fibrosis F2-F4 F2 F3 F4 |
0.66 ± 0.18 0.45±0.07 0.65±0.04 0.82±0.08 |
0.55 ± 0.18 0.43±0.13 0.51±0.09 0.67±0.16 |
0.001 0.205 0.001 0.001 |
Steatosis S0-S3 S0 S1 S2 S3 |
0.50 ± 0.18 0.33±0.14 0.34±0.08 0.58±0.06 0.76±0.05 |
0.34 ± 0.14 0.30±0.14 0.25±0.11 0.40±0.12 0.45±0.11 |
0.001 0.481 0.001 0.001 0.001 |
SVR, sustained viral response; T0, initial visit; T3, second visit
Three months post-SVR fibrosis F4 cases decreased significantly for GG genotype (from 61.5% to 38.5%) and F3 cases incresed from 30.8% to 46.2%.
Three months post-SVR steatosis S2 cases decreased significantly for GG genotype (from 53.8% to 23.1%) and S0 cases incresed from 0% to 38.5% (Table 6).
Table 6. Post-treatment fibrosis and steatosis according to PNPLA3 genotype three months post-SVR
Parameters |
CC (n=58) |
CG (n=29) |
GG (n=13) |
Test statistics |
p |
Fibrosis F1 (%) F2 (%) F3 (%) F4 (%) |
5 (8.6%) 35 (60.3%) 11 (19.0%) 7 (12.1%) |
2 (6.9%) 15 (51.7%) 8 (27.6%) 4 (13.8%) |
0 (0.0%) 2 (15.4%) 6 (46.2%) 5 (38.5%) |
Kruskal-Wallistests |
0.050 |
Steatosis S0 (%) S1 (%) S2 (%) S3 (%) |
23 (39.7%) 27 (46.6%) 8 (13.8%) 0 (0.0%) |
9 (31.0%) 16 (55.2%) 4 (13.8%) 0 (0.0%) |
5 (38.5%) 4 (30.8%) 3 (23.1%) 1 (7.7%) |
Kruskal-Wallistests |
0.035 |
Q14. In this setting, is liver steatosis only explained by HCV? Genotype 1 is not as steatogenic as genotype 3, are there other factors determining steatosis in these patients? This is a strong limitation of the study: pnpla3 is mainly related to steatosis, favoring then inflammation and thus fibrosis: this point deserves a greater attention, in my opinion.
Answer 14:
Thank you for this remark!
Indeed, we excluded patients with with diabetes mellitus, obese subjects (BMI > 29), alcohol consumers, patients with viral co-infections, and patients with steatogenic medication as major causes of steatosis. Although genotype 1 is not as steatogenic as genotype 3, it seems that HCV can cause steatosis frequently. The high incidence of steatosis in our study group might be explained by other individual factors.
Q15. No imaging available for F4 patients? How was the diagnosis made? The use of Fibromax alone would not give the appropriate certaintly that is required in this subgroup of patients.
Answer 15:
Thank you!
The diagnosis of liver cirrhosis was not based only on Fibromax but also on clinical, biological, abdominal ultrasound and upper endoscopy exams as stated in the “Materials and Methods” section: “Patients were evaluated pretreatment and three months after obtaining SVR based on clinical, biological and imaging criteria (abdominal ultrasound and upper endoscopy) at the Institute of Gastroenterology and Hepatology within the County Emergency Hospital "St. Spiridon”, Iași, from January 2018 to March 2020.”
Reviewer: 2
Answer 0.
We would like to thank the Reviewer and the Editorial board for all the remarks regarding our work. We assure the EIC that we have read carefully every suggestion from this decision letter and tried our best to improve the quality of the document accordingly.
Q1.Do the data in Table 2 include all 100 patients? Since the authors state that repeatable evaluation of fibrosis and steatosis was performed at different time points.
Answer 1:
Thank you for this question.
Yes, the data include all 100 patients. Data in Table 2 (now Table 3) are in reference to the steatosis and fibrosis degrees of all 100 patients at time T0 and T3, respectively.
Q2.Please, provide more data (mean±SD) about the patient’s cohort in respect to the PNPLA3 genotype: KMI, liver panel, and other biochemical indices used for Fibromax.
Answer 2:
Great point!
Following the reviewer’s suggestion, we added the following paragraph and table in the “Results” sectionto provide more data (mean±SD) in respect to PNPLA3 genotype (CC, CG, GG):
“The subgroup distribution of the baseline characteristics according to the PNPLA3 genotype, age, BMI, gender and the degrees of fibrosis and steatosis is illustrated in Table 2.”
Table 2. Baseline characteristics according to PNPLA3 genotype.
Parameters |
CC (n=58) |
CG (n=29) |
GG (n=13) |
Test statistics |
p |
Age Average ± SD |
61.16±9.66 |
60.24±6.61 |
60.00±7.72 |
F ANOVA test |
0.850 |
BMI Average ± SD |
27.52±4.74 |
26.43±3.23 |
27.67±6.50 |
F ANOVA test |
0.548 |
Gender Male (%) Female (%) |
20 (34.5%) 38 (65.5%) |
11 (37.9%) 18 (62.1%) |
4 (30.8%) 9 (69.2%) |
Kruskal-Wallistests |
0.896 |
Fibrosis F2 (%) F3 (%) F4 (%) |
20 (34.5%) 10 (17.2%) 28 (48.3%) |
15 (51.7%) 7 (24.1%) 7 (24.1%) |
1 (7.7%) 4 (30.8%) 8 (61.5%) |
Kruskal-Wallistests |
0.024 |
Steatosis S0 (%) S1 (%) S2 (%) S3 (%) |
9 (15.5%) 19 (32.8%) 14 (24.1%) 16 (27.6%) |
3 (10.3%) 8 (27.6%) 16 (55.2%) 2 (6.9%) |
0 (0.0%) 5 (38.5%) 7 (53.8%) 1 (7.7%) |
Kruskal-Wallistests |
0.013 |
Q3.The post-SVR means after 6 months of follow-up after treatment. If the authors write “0-3 months post-SVR” a reader should understand – 6-9 months after the end of the treatment course. The post-treatment means the end of the treatment course. In Table 2 the authors compared F and S before and after treatment, though in the text above they state that they had compared data before treatment and post-SVR. This also applies to the presentation of other data. Please, indicate clearly the time points of all tests and correct their presentation in the tables and further in the Results and Discussion sections.
Answer 3:
Thank you for this very important observation!
We apologize for the inadequacies. We indeed compared data before treatment and post-SVR and corrected allinadequaciesthroughout the manuscript (including Tables, Results and Discussions).
Q4.Table 3 data require SD and p. It would be also very useful to present post-SVR data and/or post-treatment data of S and F in the same table. It seems like there was no correlation of F and S improvement after treatment with PNPLA3 genotypes.
Answer 4:
Thank you for this important and useful suggestion.
Following the Reviewer’s observation we added SD and p data in Table 4 (that is the old Table 3). Moreover, we added post-SVR data in Table 5, as follows:
Table 4. Mean±SD values of steatosis and fibrosis before treatment according to PNPLA3 genotype (Bonferroni post hoc test pre-treatment).
|
F pre-treatment |
S pre-treatment |
||||
CC |
CG |
GG |
CC |
CG |
GG |
|
mean±SD |
0.65±0.18 |
0.60±0.17 |
0.79±0.12 |
0.51±0.20 |
0.48±0.17 |
0.52±0.16 |
CC |
- |
P=0.459 |
P=0.042 |
- |
P=0.999 |
P=0.999 |
CG |
P=0.459 |
- |
P=0.004 |
P=0.999 |
- |
P=0.999 |
GG |
P=0.042 |
P=0.004 |
- |
P=0.999 |
P=0.999 |
- |
F, fibrosis; S, steatosis
Table 5. Mean±SD values of steatosis and fibrosis after treatment (three months post-SVR) according to PNPLA3 genotype (Bonferroni post hoc test post-SVR).
|
F three months post-SVR |
S three months post-SVR |
||||
CC |
CG |
GG |
CC |
CG |
GG |
|
mean±SD |
0.52±0.17 a) |
0.55±0.18 b) |
0.70±0.15 b) |
0.33±0.14 a) |
0.35±0.13 a) |
0.37±0.18 b) |
CC |
- |
P=0.999 |
P=0.003 |
- |
P=0.999 |
P=0.916 |
CG |
P=0.999 |
- |
P=0.027 |
P=0.999 |
- |
P=0.999 |
GG |
P=0.003 |
P=0.027 |
- |
P=0.916 |
P=0.999 |
- |
F, fibrosis; S, steatosis
- a) p<0.001 for Paired samples t-test pre- vs post-SVR
- b) p<0.05 for Paired samples t-test pre- vs post-SVR
Q5. In my opinion, this article would be more correct to talk about decreasing biochemical markers of F and S because the histological improvement was not assessed. That should be stressed in the Discussion.
Answer 5:
Very good point!
Indeed, we added this aspect as a limitation in the “Discussions” section, quote:
“In addition, another limitation that should be mentioned is that the severity of the liver fibrosis and steatosis was evaluated using a noninvasive method and not by direct histological assessment.”
However, according to the latest meta-analysis comparing fibrosis biomarkers with biopsy in hepatitis C and B patients (https://pubmed.ncbi.nlm.nih.gov/26516104/), Fibrotest was reported to have very good performance. Moreover, Fibrotestis accepted and validated by all international guidelines (https://www.journal-of-hepatology.eu/article/S0168-8278(15)00259-7/pdf, https://aasldpubs.onlinelibrary.wiley.com/doi/pdf/10.1002/hep.31060) in the assessment of liver fibrosis.
Reviewer 2 Report
Do the data in Table 2 include all 100 patients? Since the authors state that repeatable evaluation of fibrosis and steatosis was performed at different time points.
Please, provide more data (mean±SD) about the patient’s cohort in respect to the PNPLA3 genotype: KMI, liver panel, and other biochemical indices used for Fibromax.
The post-SVR means after 6 months of follow-up after treatment. If the authors write “0-3 months post-SVR” a reader should understand – 6-9 months after the end of the treatment course. The post-treatment means the end of the treatment course. In Table 2 the authors compared F and S before and after treatment, though in the text above they state that they had compared data before treatment and post-SVR. This also applies to the presentation of other data. Please, indicate clearly the time points of all tests and correct their presentation in the tables and further in the Results and Discussion sections.
Table 3 data require SD and p. It would be also very useful to present post-SVR data and/or post-treatment data of S and F in the same table. It seems like there was no correlation of F and S improvement after treatment with PNPLA3 genotypes.
In my opinion, this article would be more correct to talk about decreasing biochemical markers of F and S because the histological improvement was not assessed. That should be stressed in the Discussion.
Author Response

(The authors gave the same response as above.)

Round 2
Reviewer 1 Report
The Authors have addressed all points and the manuscript is clearer and improved. In this form it appears to be suitable for publication.
Author Response
Dear Editor,
We are grateful to the Editorial Board for considering a revised version of our manuscript entitled “Role of PNPLA3 in the Assessment and Monitoring of Hepatic Steatosis and Fibrosis in Patients with Chronic Hepatitis C Infection Who Achieved a Sustained Virologic Response” in the Medicina.
We would like to thank the reviewers for their valuable time and useful contribution. We much appreciate their input, which helped improve our manuscript.
Also, we look forward to hearing from you regarding our submission. We would be glad to respond to any further questions and comments that you may have.
Please find below a detailed point-by-point reply to the comments made by reviewers.
Sincerely,
--
Oana Irina Gavril, MD, PhD
University of Medicine and Pharmacy “Gr. T. Popa” - Iasi - Romania
On behalf of the authors
Letter to the Editor
Reviewers’ Comments to Authors:
Reviewer: 2
Q1. In present version the manuscript sounds better, but still it needs to be improved. Please, add following data (average+-SD and p) in the Table 2: ALT, AST, gamaGT, ALP, total bilirubin, apolipoprotein A1, alfa-2-macroglobulin, haptoglobulin, cholesterol, triglicerides, fasting glucose.
Answer 1.
Great point!
Following the reviewer’s suggestion, we added the requested data in “Table 2”.
“Before treatment, the mean levels of total bilirubin were significantly increased in GG genotype compared to CC and CG genotypes (1.90 vs. 0.77 and 0.94; p=0.001) and the mean levels of total cholesterol were significantly increased in CC genotype compared to CG and GG genotypes (154.14 vs. 147.62 and 115.33; p=0.019), the others parameters, except aspartate aminotransferase (AST), registering average values slightly lower for the GG genotype than those registered for the CC or CG genotypes (p>0.05).”
Table 2. Baseline characteristics according to PNPLA3 genotype.
Parameters |
CC (n=58) |
CG (n=29) |
GG (n=13) |
Test statistics |
p |
Age Average ± SD |
61.16±9.66 |
60.24±6.61 |
60.00±7.72 |
F ANOVA test |
0.850 |
BMI Average ± SD |
27.52±4.74 |
26.43±3.23 |
27.67±6.50 |
F ANOVA test |
0.548 |
Gender Male (%) Female (%) |
20 (34.5%) 38 (65.5%) |
11 (37.9%) 18 (62.1%) |
4 (30.8%) 9 (69.2%) |
Kruskal-Wallistests |
0.896 |
Fibrosis F2 (%) F3 (%) F4 (%) |
20 (34.5%) 10 (17.2%) 28 (48.3%) |
15 (51.7%) 7 (24.1%) 7 (24.1%) |
1 (7.7%) 4 (30.8%) 8 (61.5%) |
Kruskal-Wallistests |
0.024 |
Steatosis S0 (%) S1 (%) S2 (%) S3 (%) |
9 (15.5%) 19 (32.8%) 14 (24.1%) 16 (27.6%) |
3 (10.3%) 8 (27.6%) 16 (55.2%) 2 (6.9%) |
0 (0.0%) 5 (38.5%) 7 (53.8%) 1 (7.7%) |
Kruskal-Wallistests |
0.013 |
ALT Average ± SD |
94.50±64.24 |
87.92±63.18 |
82.17±51.02 |
F ANOVA test |
0.878 |
AST Average ± SD |
69.02±36.66 |
68.69±44.00 |
91.00±39.82 |
F ANOVA test |
0.422 |
GammaGT Average ± SD |
63.67±42.59 |
66.00±49.69 |
35.00±16.81 |
F ANOVA test |
0.286 |
ALP Average ± SD |
84.86±22.55 |
103.80±39.70 |
89.69±11.50 |
F ANOVA test |
0.114 |
Totalbilirubin Average ± SD |
0.77±0.33 |
0.94±0.63 |
1.90±1.29 |
F ANOVA test |
0.001 |
ApolipoproteinA1 Average ± SD |
1.43±0.29 |
1.36±0.27 |
1.18±0.29 |
F ANOVA test |
0.129 |
Alfa-2-macroglobulin Average ± SD |
3.52±0.56 |
3.18±0.70 |
3.00±0.85 |
F ANOVA test |
0.072 |
Heptoglobulin Average ± SD |
0.72±0.34 |
0.85±0.57 |
0.40±0.29 |
F ANOVA test |
0.080 |
TotalCholesterol Average ± SD |
154.14±27.78 |
147.62±35.13 |
115.33±34.72 |
F ANOVA test |
0.019 |
Triglicerides Average ± SD |
90.49±32.70 |
97.15±24.37 |
85.50±18.64 |
F ANOVA test |
0.689 |
Fasting glucose Average ± SD |
99.54±9.94 |
97.85±9.56 |
90.00±9.42 |
F ANOVA test |
0.094 |
F, fibrosis; S, steatosis; BMI, body mass index; ALT, alanine aminotransferase; AST, aspartate aminotransferase; GGT, gamma-glutamyl transferase; ALP, alkaline phosphatase
Q2. In the study limitation section, 6 months of patients' monitoring time is indicated, but in the manuscript text the authors have compared pre-treatment data and three months post-SVR data. Taking into account SVR time point, it looks like the authors have presented post-SVR data. Please, specify.
Answer 2.
Thank you for this important point. We updated the text as it follows:
“Our study has a series of limitations that must be mentioned. One of these is the limited number of patients and the short monitoring period (6 months).”
…was replaced by:
“Our study has a series of limitations that must be mentioned. One of these is the limited number of patients and the short monitoring period (9 months).”

Reviewer 2 Report
In present version the manuscript sounds better, but still it needs to be improved. Please, add following data (average+-SD and p) in the Table 2: ALT, AST, gamaGT, ALP, total bilirubin, apolipoprotein A1, alfa-2-macroglobulin, haptoglobulin, cholesterol, triglicerides, fasting glucose.
In the study limitation section, 6 months of patients' monitoring time is indicated, but in the manuscript text the authors have compared pre-treatment data and three months post-SVR data. Taking into account SVR time point, it looks like the authors have presented post-SVR data. Please, specify.
Author Response

(The authors gave the same response as above.)
